# Counterfactually Fair Representation

**Zhiqun Zuo**[1] **Mohammad Mahdi Khalili**[1,2] **Xueru Zhang**[1]
zuo.167@osu.edu      khalili.17@osu.edu      zhang.12807@osu.edu

[1]CSE Department, The Ohio State University, Columbus, OH 43210
[2]Yahoo Research, New York, NY, 10003

## Abstract

The use of machine learning models in high-stake applications (e.g., healthcare, lending, college admission) has raised growing concerns due to potential biases against protected social groups. Various fairness notions and methods have been proposed to mitigate such biases. In this work, we focus on Counterfactual Fairness (CF), a fairness notion that is dependent on an underlying causal graph and first proposed by Kusner *et al.* [26]; it requires that the outcome an individual perceives is the same in the real world as it would be in a "counterfactual" world, in which the individual belongs to another social group. Learning fair models satisfying CF can be challenging. It was shown in [26] that a sufficient condition for satisfying CF is to **not** use features that are descendants of sensitive attributes in the causal graph. This implies a simple method that learns CF models only using non-descendants of sensitive attributes while eliminating all descendants. Although several subsequent works proposed methods that use all features for training CF models, there is no theoretical guarantee that they can satisfy CF. In contrast, this work proposes a new algorithm that trains models using all the available features. We theoretically and empirically show that models trained with this method can satisfy CF[1].

## 1 Introduction

While machine learning (ML) has had significant impacts on human-involved applications (e.g., lending, hiring, healthcare, criminal justice, college admission), it also poses significant risks, particularly regarding unfairness against protected social groups. For example, it has been shown that computer-aided clinical diagnostic systems can exhibit discrimination against people of color [9]; face recognition surveillance technology used by police may have racial bias [1]; a decision support tool COMPAS used for predicting the recidivism risk of defendants is biased against African Americans [5]. Various fairness notions have been proposed to mathematically measure the biases in ML based on observational data. Examples include: i) *unawareness* which prohibits the use of sensitive attribute in model training process; ii) *parity-based fairness* that requires certain statistical measures to be equalized across different groups, e.g., equalized odds [20], equal opportunity [20], statistical parity [13], predictive parity [20]; iii) *preference-based fairness* that is inspired by the fair-division and envy-freeness literature in economics, it ensures that given the choice between various decision outcomes, every group of users would collectively prefer its perceived outcomes, regardless of the (dis)parity compared to the other groups [43, 12].

However, the fairness notions mentioned above do not take into account the causal structure and relations among different features/variables. Recently, Kusner *et al.* [26] proposed a fairness notion called *Counterfactual Fairness* (CF) based on the causal model and counterfactual inference; it

---

[1]The code repository for this work can be found in `https://github.com/osu-srml/CF_Representation_Learning`

requires that the ML outcome received by an individual should be the same in the real world as it would be in a counterfactual world, in which the individual belongs to a different social group. To satisfy CF, [26] shows that it is sufficient to make predictions only using the features that are non-descendants of the sensitive attribute node in the causal graph. However, this approach may discard crucial data, as descendants of sensitive attribute may contain useful information that is critical for prediction and downstream tasks. In this work, we show that using only non-descendants is not a necessary condition for achieving CF. In particular, we propose a novel method for generating counterfactually fair representations using all available features (including both descendants and non-descendants of sensitive attribute). The idea is to first generate counterfactual samples of each data point based on the causal structure, and the fair representations can be generated subsequently by applying a *symmetric function* to both factual (i.e., original data) and counterfactual samples. We can theoretically show that ML models (or any other downstream tasks) trained with counterfactually fair representations can satisfy *perfect* CF. Experiments on real data further validate our theorem.

It is worth noting that several subsequent studies of [26] also proposed methods to learn CF models using all available features, e.g., [14, 36, 24, 10, 2]. However, these methods are empirical and there is no theoretical guarantee that these methods can satisfy (perfect) CF. In Appendix A, we introduce more related work and discuss the differences with ours. Our main contributions are as follows:

- We propose a novel and efficient method for generating counterfactually fair representations. We theoretically show that ML models trained with such representations can achieve perfect/exact CF.

- We extend our method to path-dependent counterfactual fairness [26]. That is, for any unfair path in a causal graph, we can generate representations that mitigate the impact of sensitive attributes on the prediction along the unfair path.

- We conduct extensive experiments (across different causal models, datasets, and fairness definitions) to compare our method with existing methods. Empirical results show that 1) our method outperforms the method of only using non-descendants of sensitive attributes; 2) existing heuristic methods for training ML model under CF fall short of achieving perfect CF fairness.

## 2 Problem Formulation

Consider a supervised learning problem where the training dataset consists of triples $V = (X, A, Y)$, where random vector $X = [X_1, \cdots, X_d]^\top \in \mathcal{X}$ are observable features, $A \in \mathcal{A}$ is the sensitive attribute (e.g., race, gender) indicating the group membership, and $Y \in \mathcal{Y} \subseteq \mathbb{R}$ is the label/output. Similar to [31], we associate the observable $V = (X, A, Y)$ with a causal model $\mathcal{M} = (U, V, F)$, where $U$ is a set of unobserved (exogenous) random variables that are factors not caused by any variable in $V$, and $F = \{f_1, f_2, \cdots, f_d, f_{d+1}, f_{d+2}\}$ is a set of functions (a.k.a. structural equations [4]) with one for each variable in $V$. WLOG, let

$$X_i = f_i(pa_i, U_{pa_i}), \ i \in \{1, \cdots, d\}; \quad A = f_{d+1}(pa_{d+1}, U_{pa_{d+1}}); \quad Y = f_{d+2}(pa_{d+2}, U_{pa_{d+2}}),$$

where $pa_i$ and $U_{pa_i}$ are the sets of observable and unobservable variables that are the parents of $X_i$. $pa_{d+1}$ and $pa_{d+2}$ (resp. $U_{pa_{d+1}}$ and $U_{pa_{d+2}}$) are the observable (resp. unobservable) variables that are parents of $A$ and $Y$, respectively. Assume $(U, V)$ can be represented as a directed acyclic graph.

Our goal is to learn a predictor $\hat{Y} = g_w(R)$ parameterized by weight vector $w \in \mathbb{R}^{d_w}$ from training data. Here $R = h(X, A; \mathcal{M})$ is a representation generated using $(X, A)$ and causal model $\mathcal{M}$. Define loss function $l : \mathcal{Y} \times \mathbb{R} \to \mathbb{R}$ where $l(Y, g_w(R))$ is the loss associated with $g_w$ in estimating $Y$ using representation $R$. We denote the expected loss with respect to the joint probability distribution of $(R, Y)$ by $L(w) := \mathbb{E}\{l(Y, g_w(R))\}$. Throughout the paper, we use small letters to denote the realizations of random variables, e.g., $(x, a, y)$ is a realization of $(X, A, Y)$.

### 2.1 Background: Intervention and Counterfactual Inference

Given structural equations and the distribution of unobservable variables $U$, we can calculate the distribution of any observed variable $V_i \in V$ and even study the impact of intervening certain observed variables on other variables. Specifically, the **intervention** on variable $V_i$ can be done by simply replacing structural equation $V_i = f_i(pa_i, U_{pa_i})$ with equation $V_i = v$ for some $v$. To study the impact of intervening $V_i$, we can use new structural equations to find resulting distributions of other observable variables and see how they may differ as $v$ changes.

The specification of structural equations $F$ further allows us to compute **counterfactual** quantities, i.e., computing the value of $Y$ if $Z$ had taken value $z$ for two observable variables $Z, Y$. Because the value of any observable variable is fully determined by unobserved variables $U$ and structural equations, the counterfactual value of $Y$ for a given $U = u$ can be computed by replacing structural equations for $Z$ as $Z = z$. Such counterfactual value is typically denoted as $Y_{Z \leftarrow z}(u)$.

The goal of **counterfactual inference** is to compute the probabilities $\Pr\{Y_{Z \leftarrow z}(U)|O = o\}$ for some observable variables $O$. It can be used to infer "the value of $Y$ if $Z$ had taken value $z$ in the presence of evidence $O = o$". Based on [16], $\Pr\{Y_{Z \leftarrow z}(U)|O = o\}$ can be computed in three steps: (i) *abduction* that finds posterior distribution of $U$ given $O = o$ for a given prior on $U$; (ii) *action* that performs intervention $Z = z$ by replacing structural equations of $Z$; (iii) *prediction* that computes the distribution of $Y$ using new structural equations and the posterior $\Pr\{U|O = o\}$

## 2.2 Counterfactual Fairness

Without fairness consideration, simply learning a predictor by minimizing the expected loss, i.e., $\arg\min_w L(w)$, may exhibit biases against certain social groups. One way to tackle unfairness issue is to enforce a certain fairness constraint when learning the predictor. In this work, we consider *counterfactual fairness* as formally defined below.

**Definition 1** (Counterfactual Fairness (CF) [26])**.** *We say a predictor $\hat{Y} = g_w(R)$ satisfies CF if the following holds for every $(x, a)$:*

$$\Pr\{\hat{Y}_{A \leftarrow a}(U) = y|X = x, A = a\} = \Pr\{\hat{Y}_{A \leftarrow a'}(U) = y|X = x, A = a\}, \; \forall y \in \mathcal{Y}, a' \in \mathcal{A}.$$

This notion suggests that any intervention on sensitive attribute $A$ should not change the distribution $\hat{Y}$ given that $U$ follows distribution $\Pr_{\mathcal{M}}\{U|X = x, A = a\}^2$. In other words, adjusting $A$ should not affect the distribution of $\hat{Y}$ if we keep other factors that are not causally dependent on $A$ constant. Learning a fair predictor satisfying CF can be challenging. As shown in [26], *a sufficient condition for satisfying CF is to **not** use features that are descendant of $A$*. In other words, given training dataset $D = \{x^{(i)}, a^{(i)}, y^{(i)}\}_{i=1}^n$, it suffices to minimize the following empirical risk to satisfy CF:

$$\arg\min_w \frac{1}{n} \sum_{i=1}^n \mathbb{E}\left\{l(y^{(i)}, g_w(U^{(i)}, x^{(i)}_{\not\prec A}))|X = x^{(i)}, A = a^{(i)}\right\},$$

where $x^{(i)}_{\not\prec A}$ are non-descendant features of $A$ corresponding to $i$-th sample, and the expectation is with respect to the random variable $U^{(i)} \sim \Pr_{\mathcal{M}}\{U|X = x^{(i)}, A = a^{(i)}\}$.

Although removing the descendants of $A$ from the input is a simple method to address the unfairness issue, it comes at the cost of losing important information. In some examples (e.g., the ones provided in [26]), it is possible that all (or most of) the features are descendants of $A$ and need to be eliminated when training the predictor. We thus ask:

*Can we train a predictor that satisfies **perfect** CF using all the available features as input?*

Although several recently proposed methods try to train CF predictors using all the available features (including both non-descendants and descendants of $A$) [14, 36], there is no guarantee that they can satisfy CF. In contrast, our work aims to propose a theoretically-certified algorithm that finds counterfactually fair predictors using all the available features.

## 3 Proposed Method

In this section, we introduce our algorithm for training a supervised model under CF. Our method consists of three steps: (i) counterfactual samples generation; (ii) counterfactually fair representation generation; and (iii) fair model training. We present each step in detail as follows.

**1. Counterfactual samples generation.** We first introduce the definition of counterfactual samples and then present the method for generating them. They will be used for generating CF representations.

---

[2]Sometimes, we use subscript $\mathcal{M}$ to emphasize that the distribution is calculated based on causal model $\mathcal{M}$ which we assume is known.

**Definition 2** (Counterfactual Sample). *Consider $i$-th data point in training dataset with feature vector $x^{(i)}$ and sensitive attribute $a^{(i)}$. Let $u^{(i)}$ be the unobservable variable associated with $(x^{(i)}, a^{(i)})$ sampled from distribution $\mathrm{Pr}_\mathcal{M}\{U|X = x^{(i)}, A = a^{(i)}\}$ under causal model $\mathcal{M} = (V, U, F)$. Then, $(\check{x}^{(i)}, \check{a}^{(i)})$ is a counterfactual sample with respect to $(x^{(i)}, a^{(i)})$ if $\check{a}^{(i)} \neq a^{(i)}$ and $\check{x}^{(i)}$ is generated using structural equations $F$, unobservable variable $U = u^{(i)}$, and intervention $A = \check{a}^{(i)}$.[3]*

Equivalently, we can represent the counterfactual feature $\check{x}^{(i)} = \check{X}_{A\leftarrow \check{a}^{(i)}}^{(i)}(u^{(i)})$. Next, we use an example to clarify the generation process of counterfactual sample.

**Example 1** (**Law School Success** [26]). *Consider a group of students, each has observable features grade-point average (GPA) before entering college $X_G$ and entrance exam score (LSAT) $X_L$. Let first-year average grade in college (FYA) be label $Y$ to be predicted and let race $Q$ and sex $S$ be the sensitive attributes. Suppose there are three unobservable variables $U_G, U_L, U_F$ representing errors and the relations between these variables can be characterized by the following structural equations:*

$$
\begin{aligned}
X_G &= GPA = b_G + w_G^Q Q + w_G^S S + U_G, \\
X_L &= LSAT = b_L + w_L^Q Q + w_L^S S + U_L, \\
Y &= FYA = b_F + w_F^Q Q + w_F^S S + U_F,
\end{aligned}
$$

*where $(b_G, b_L, b_F, w_G^Q, w_G^S, w_L^Q, w_L^S, w_F^R, w_F^S)$ are the parameters of the causal model, which we assume are given[4]. Consider one student with $x^{(0)} = (x_G^{(0)}, x_L^{(0)})$ and $a^{(0)} = (q^{(0)}, s^{(0)})$. To generate its counterfactual sample, we first compute the underlying unobservable variables $(u_G^{(0)}, u_L^{(0)})$:*

$$
\left( u_G^{(0)}, u_L^{(0)} \right) = \left( x_G^{(0)} - b_G - w_G^Q q^{(0)} - w_G^S s^{(0)}, \; x_L^{(0)} - b_L - w_L^Q q^{(0)} - w_L^S s^{(0)} \right).
$$

*Then, for any $(\check{q}, \check{s}) \in \mathcal{A} - \{(q^{(0)}, s^{(0)})\}$, the corresponding counterfactual features can be generated:*

$$
\begin{aligned}
\check{x}_G^{(0)} &= b_G + w_G^Q \check{q} + w_G^S \check{s} + u_G^{(0)} = x_G^{(0)} + w_G^Q (\check{q} - q^{(0)}) + w_G^S (\check{s} - s^{(0)}) \\
\check{x}_L^{(0)} &= b_L + w_L^Q \check{q} + w_L^S \check{s} + u_L^{(0)} = x_L^{(0)} + w_L^Q (\check{q} - q^{(0)}) + w_L^S (\check{s} - s^{(0)})
\end{aligned}
$$

In the above example, finding unobservable variables is straightforward due to the additive error model (i.e., each observable variable $V_i$ is equal to $f_i(pa_i) + U_i$ ). For other causal models that are non-additive, we can leverage techniques such as Variational Auto Encoder (VAE) to first learn distribution $\mathrm{Pr}_\mathcal{M}\{U|X = x, A = a\}$ and then sample from this distribution, see e.g., [21, 24, 30, 37].

**2. Counterfactually fair representation generation.** Next, we introduce how to generate counterfactually fair representation $R = h(X, A; \mathcal{M}, s)$ using counterfactual samples generated above.

The complete procedure is given in Algorithm 1. The idea is to first apply a *symmetric function* $s(\cdot)$ to both factual feature $x$ and counterfactual features $\{\check{x}^{[j]}\}_{j=1}^{|\mathcal{A}|-1}$. This output can be leveraged to generate CF representation. The symmetry of the function is formally defined below.

**Definition 3.** *A function $s : \mathcal{X}^{|\mathcal{A}|} \to \mathbb{R}$ is symmetric if the output is the same for any permutation of inputs.*

One example of symmetric function is the average over all inputs, e.g., $s(x, \check{x}^{[1]}, \ldots, \check{x}^{[|A|-1]}) = \frac{(x + \check{x}^{[1]} + \ldots + \check{x}^{[|A|-1]})}{|A|}$.

---

**Algorithm 1** CF Representation Generation $h(x, a; \mathcal{M}, s)$

**Input:** Causal model $\mathcal{M}$, observable features $(x, a)$, symmetric function $s$

1: Sample $u$ from distribution $\mathrm{Pr}_\mathcal{M}\{U|X = x, A = a\}$
2: Use $u$ and causal model $\mathcal{M}$ to generate $|\mathcal{A}| - 1$ counterfactual samples $\{(\check{x}^{[1]}, \check{a}^{[1]}), \ldots, (\check{x}^{[|\mathcal{A}|-1]}, \check{a}^{[|\mathcal{A}|-1]})\}$, where $\check{a}^{[j]} \in \mathcal{A} - \{a\}$.
3: Use symmetric function $s(.)$ to generate representation

$$R = [s(x, \check{x}^{[1]}, \ldots, \check{x}^{[|\mathcal{A}|-1]}), u]$$

**Output:** counterfactually fair representation $R$

---

**3. Fair model training.** Given CF representation $R = h(X, A; \mathcal{M}, s)$ generated by Algorithm 1, we can use it directly to learn a predictor that satisfies CF. Indeed, we can show that any predictor learned based on CF representation satisfies perfect CF, as stated in Theorem 1 below.

---

[3]If $A$ is non-binary, we can generate $|\mathcal{A}| - 1$ counterfactual samples for each $\check{a} \in \mathcal{A} - \{a\}$. We can use $\check{x}^{[j]}$ to represent $j$-th counterfactual sample corresponding to $x$ and the $j$-th element in $\mathcal{A} - \{a\}$.

[4]The parameters of a causal model can be found using observational data and by the maximum likelihood estimation. As a result, we can assume that the parameters of the causal model are given.

**Theorem 1.** *If representation is generated based on $h(x, a; \mathcal{M}, s)$ in Algorithm 1, then the predictor $g_w(h(x, a; \mathcal{M}, s))$ satisfies perfect CF for all $w \in \mathbb{R}^{d_w}$.*

Because $g_w(h(x, a; \mathcal{M}, s))$ satisfies CF for all parameter $w$, we can find the optimal predictor directly by solving an unconstrained optimization:

$$w^* = \arg\min_w \frac{1}{n} \sum_{i=1}^{n} l\left(y^{(i)}, g_w(h(x^{(i)}, a^{(i)}; \mathcal{M}, s))\right)$$

Under Theorem 1, it is guaranteed that the optimal predictor $g_{w^*}$ satisfies counterfactual fairness.

**Inference.** After learning the optimal CF predictor $g_{w^*}$, we can use it to make fair predictions about new data. At the inference phase, for a given example $(x, a)$, we first generate its CF representation using Algorithm 1 and find the prediction $\hat{y}$ using $g_{w^*}$. That is, $\hat{y} = g_{w^*}(h(x, a; \mathcal{M}, s))$.

**Discussion.** Compared to [26], our method leverages all available features and can attain much better performance without sacrificing fairness. We will further validate this in experiments (Section 5). As mentioned earlier, there are existing methods that also use all features to train predictors under CF constraint. For instance, the method proposed in [24] also generates the counterfactual sample for each training data and it trains model using both factual and counterfactual samples; [14] learns CF predictor by adding a penalty term to the learning objective function, where the penalty term is calculated based on the counterfactual samples. While these two methods also leverage counterfactual samples to reduce bias, they cannot satisfy *perfect* CF and there is no theoretical guarantee.

## 4 Path-dependent Counterfactual Fairness

In this section, we consider a variant notion of CF called *Path-dependent Counterfactual Fairness* (PCF). We will show how the proposed method can be adapted to train predictors under PCF. Let $\mathcal{G}$ be the graph associated with causal model $\mathcal{M}$, and $\mathcal{P}_{\mathcal{G}_A}$ be the set of all unfair directed paths from sensitive attribute $A$ to output $Y$. Further, we define $X_{\mathcal{P}_{\mathcal{G}_A}^c}$ as the features that are not present in any unfair path in $\mathcal{P}_{\mathcal{G}_A}$ and $X_{\mathcal{P}_{\mathcal{G}_A}}$ as the features along the unfair paths. Path-dependent Counterfactual Fairness is defined as follows.

**Definition 4** (Path-dependent Counterfactual Fairness (PCF) [26])**.** *We say $\hat{Y} = g_w(R)$ satisfies PCF with respect to path set $\mathcal{P}_{\mathcal{G}_A}$ if the following holds for every $(x, a)$: $\forall y \in \mathcal{Y}, a' \in \mathcal{A}$*

$$\Pr\{\hat{Y}_{A \leftarrow a, X_{\mathcal{P}_{\mathcal{G}_A}^c} \leftarrow x_{\mathcal{P}_{\mathcal{G}_A}^c}}(U) = y | X = x, A = a\} = \Pr\{\hat{Y}_{A \leftarrow a', X_{\mathcal{P}_{\mathcal{G}_A}^c} \leftarrow x_{\mathcal{P}_{\mathcal{G}_A}^c}}(U) = y | X = x, A = a\}$$

This notion suggests that if we fix attributes $X_{\mathcal{P}_{\mathcal{G}_A}^c}$ and let $U$ follows posterior distribution $\Pr_{\mathcal{M}}\{U | X = x, A = a\}$, then variable $A$ should not affect the predictor along unfair path(s) in $\mathcal{P}_{\mathcal{G}_A}$. Note that PCF reduces to CF if $X_{\mathcal{P}_{\mathcal{G}_A}^c} = \emptyset$. We want to emphasize that path-*specific* counterfactual fairness defined in [7] is different from Definition 4. Path-*specific* counterfactual fairness [7] considers a baseline value $a'$ for $A$, and requires $A = a'$ propagate

**Algorithm 2** PCF Representation Generation $h(x, a; \mathcal{M}, s, X_{\mathcal{P}_{\mathcal{G}_A}^c})$

---

**Input:** Causal model $\mathcal{M}$, Set of variables $X_{\mathcal{P}_{\mathcal{G}_A}^c}$ that are not along unfair paths, $(x, a)$, symmetric function $s$

1: Sample $u$ from distribution $\Pr_{\mathcal{M}}\{U | X = x, A = a\}$
2: Apply intervention $X_{\mathcal{P}_{\mathcal{G}_A}^c} = x_{\mathcal{P}_{\mathcal{G}_A}^c}$ to get a new model $\mathcal{M}'$.
3: Use $u$ and causal model $\mathcal{M}'$ to generate $|\mathcal{A}| - 1$ counterfactual features $\{(\check{x}_{\mathcal{P}_{\mathcal{G}_A}}^{[1]}, \check{a}^{[1]}), \ldots, (\check{x}_{\mathcal{P}_{\mathcal{G}_A}}^{[|\mathcal{A}|-1]}, \check{a}^{[|\mathcal{A}|-1]})\}$, where $\check{a}^{[j]} \in \mathcal{A} - \{a\}$.
4: Use symmetric function $s(.)$ to generate representation

$$R = \left[x_{\mathcal{P}_{\mathcal{G}_A}^c}, s(x_{\mathcal{P}_{\mathcal{G}_A}}, \check{x}_{\mathcal{P}_{\mathcal{G}_A}}^{[1]}, \ldots, \check{x}_{\mathcal{P}_{\mathcal{G}_A}}^{[|\mathcal{A}|-1]}), u\right]$$

**Output:** path-dependent counterfactually fair representation $R$

---

through unfair paths, and the true value of $A$ propagates through other paths. In contrast, in path-*dependent* CF, there is no baseline value for $A$, and $A$ should not cause $Y$ through unfair paths.

In this section, we describe how our proposed method could be adapted to the path-dependent case. Before describing the algorithm formally, we present an example.

**Example 2** (**Representation for PCF**)**.** *Consider a causal graph shown in Figure 1. In this graph, there are two directed paths from $A$ to $Y$. We assume that $A$ is binary, and $\mathcal{P}_{\mathcal{G}_A} = \{(A \rightarrow X_2 \rightarrow Y)\}$,*

*and $X_{\mathcal{P}_{\mathcal{G}_A}^c} = \{X_1\}$. Based on Definition 4, given sample $(x, a)$, after intervention $X_1 = x_1$ (Figure 2), intervention on $A$ should not affect the prediction outcome. We generate a representation in the following steps, 1) we find distribution $\Pr_{\mathcal{M}}\{U|X = x, A = a\}$ and sample $u$ from this distribution. 2) Using the graph in Figure 2, we generate counterfactual value for $X_2$. That is, for a given structural equation $X_2 = f_2(A, X_1, U)$, we generate counterfactual value $\tilde{x}_2 = f_2(a', x_1, u)$, where $a' \neq a$. 3) We generate representation $R = [x_1, s(x_2, \tilde{x}_2), u]$, where $s$ is a symmetric function.*

Based on the above example, the representation should be generated based on causal graph $\mathcal{G}$ after intervention $X_{\mathcal{P}_{\mathcal{G}_A}^c} = x_{\mathcal{P}_{\mathcal{G}_A}^c}$. The detailed representation generation procedure under PCF is stated in Algorithm 2. Let $h(x, a; \mathcal{M}, s, X_{\mathcal{P}_{\mathcal{G}_A}^c})$ be the function that generates such a presentation using Algorithm 2. We have the following theorem for $h(x, a; \mathcal{M}, s, X_{\mathcal{P}_{\mathcal{G}_A}^c})$.

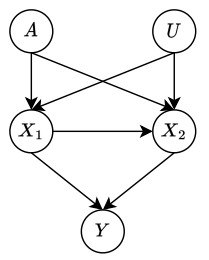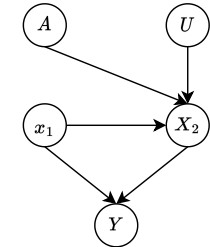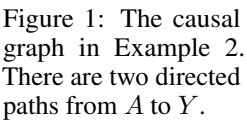

Figure 1: The causal graph in Example 2. There are two directed paths from $A$ to $Y$.

Figure 2: The causal graph in Example 2 after applying intervention $X_{\mathcal{P}_{\mathcal{G}_A}^c} = x_{\mathcal{P}_{\mathcal{G}_A}^c}$

**Theorem 2.** *Assume $R = h(x, a; \mathcal{M}, s, X_{\mathcal{P}_{\mathcal{G}_A}^c})$ is representation generated based on Algorithm 2. Then the predictor $g_w(h(x, a; \mathcal{M}, s, X_{\mathcal{P}_{\mathcal{G}_A}^c}))$ satisfies perfect PCF for all $w \in \mathbb{R}^{d_w}$.*

The above theorem implies if we train a predictor using $\{r^{(i)} = h(x^{(i)}, a^{(i)}; \mathcal{M}, s, X_{\mathcal{P}_{\mathcal{G}_A}^c}), y^{(i)}\}_{i=1}^n$, and we use $r = h(x, a; \mathcal{M}, s, X_{\mathcal{P}_{\mathcal{G}_A}^c})$ at the time of inference, then PCF is satisfied by the predictor.

## 5 Experiment

**Datasets and Causal Models.** We use the Law School Success dataset [40] and the UCI Adult Income Dataset [25] to evaluate our proposed method. The Law School Success dataset consists of 21,790 students across 163 law schools in the United States. It includes five attributes for each student: entrance exam score (LSAT), grade-point average (GPA), first-year grade (FYA), race, and gender. In our experiment, gender is the sensitive attribute $A$, and the goal is to predict FYA (label $Y$) using LSAT, GPA, Race (three features $X$), and the sensitive attribute $A$.

The UCI Adult Income Dataset contains 65,123 data instances, each with 14 attributes: age, work class, education, marital status, occupation, relationship, race, sex, hours per week, native country, and income. In our experiments, we consider sex as the sensitive attribute $A$, whether income is greater than \$50$K$ or not as the target variable $Y$, and all other attributes as features $X$.

We evaluated our methods using two distinct causal graphs. The first graph is the one presented in CVAE paper [37] (see Figure 3). We divided the feature attributes into two subsets, $X_\alpha$ and $X_\beta$. $X_\alpha$ comprises attributes that are not causally affected by $A$, while $X_\beta$ includes the remaining attributes. Exogenous variable $U$ is defined as the latent influencing factors. For the Law School Success dataset, $X_\alpha$ consists of {Race} and $X_\beta$ in-

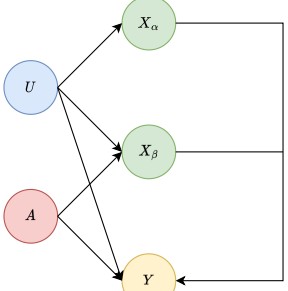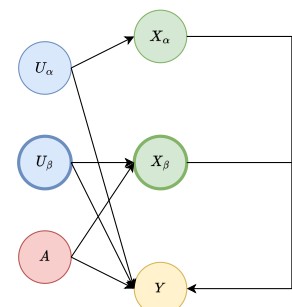

Figure 3: CVAE causal graph    Figure 4: DCEVAE causal graph

cludes {LSAT, GPA}. For the UCI Adult Income dataset, similar to [24], we assume that $X_\alpha$ contains {Age, Race, Native Country} and $X_\beta$ includes {Workclass, Education, Marital Status, Occupation, Relationship, Hours per Week}.

The second graph is proposed by DCEVAE paper [24] (see Figure 4). The main assumption in [24] is that the exogenous variables controlling $X_\alpha$ and $X_\beta$ can be disentangled into $U_\alpha$ and $U_\beta$. We used the same sets of $X_\alpha$ and $X_\beta$ as in the CVAE graph.

In contrast to the structural equations used in [26], we let structural functions be a family of functions represented by the decoder of a VAE network, as proposed in CVAE [37] and DCEVAE [24]. Moreover, the encoder is able to find unobserved variable $U$ for any given data point.[5] The parameters of the VAE network are learned from observational data. In the CVAE causal model, the decoder part includes functions $f_\alpha$, $f_\beta$, and $f_Y$ and is trained such that the following structural equations hold,

$$X_\alpha = f_\alpha(U); \ \ X_\beta = f_\beta(U, A); \ \ Y = f_Y(U, A) \tag{1}$$

For the DCEVAE causal model, the decoder part also includes functions $f_\alpha$, $f_\beta$, and $f_Y$ but it is trained such that the following holds,

$$X_\alpha = f_\alpha(U_\alpha); X_\beta = f_\beta(U_\beta, A); Y = f_Y(U_\alpha, U_\beta, A) \tag{2}$$

**Baselines and Experimental Setup.** For each dataset, we perform two separate experiments; one under CVAE causal model, and the other under DCEVAE causal model. For each experiment, we consider the following five baselines.

- **Unfair (UF).** This method trains a supervised model (logistic regression for the UCI Adult Income dataset, and linear regression for the Law School Success dataset) without any fairness constraint.
- **Counterfactual Augmentation (CA) [24].** For each sample $(x^{(i)}, y^{(i)}, a^{(i)}) \in D$, first we use the encoder of VAE to find unobserved variable $u^{(i)}$. Then, we use the decoder to generate both factual and counterfactual samples. Specifically, for counterfactual samples, we use $u^{(i)}$ and $\check{a}^{(i)[1]} \neq a^{(i)}$ as the input of decoder to generate $\check{x}^{(i)[1]}$ and $\check{y}^{(i)[1]}$. For factual data, we use $u^{(i)}$ and $\check{a}^{(i)[0]} = a^{(i)}$ as the input of the decoder to generate $\check{x}^{(i)[0]}$ and $\check{y}^{(i)[0]}$. We use the following dataset to train a predictor, $\check{D} = \{(\check{x}^{(i)[0]}, \check{a}^{(i)[0]}, \check{y}^{(i)[0]}), (\check{x}^{(i)[1]}, \check{a}^{(i)[1]}, \check{y}^{(i)[1]}) | i = 1, \ldots, n\}$.
- **Improved Counterfactual Augmentation (ICA).** We realized that the baseline CA could further be improved by training the predictor using $\check{D} = \{(x^{(i)}, a^{(i)}, y^{(i)}), (\check{x}^{(i)[1]}, \check{a}^{(i)[0]}, \check{y}^{(i)[1]}) | i = 1, \ldots, n\}$. Thus, this improved method is also considered as a baseline in experiments.
- **Counterfactual Training Using Exogenous Variable (CE) [26].** This method only uses the variables that are not descendants of the sensitive attribute.
- **Counterfactual Regularizer (CR) [14].** This baseline adds a regularizer $||\hat{y}^{(i)} - \check{y}^{(i)[1]}||_2$ to the loss function, where $\hat{y}^{(i)}$ is the output of the predictor for input $(x^{(i)}, a^{(i)})$ and $\check{y}^{(i)[1]}$ is the output of the prediction for $(\check{x}^{(i)[1]}, \check{a}^{(i)[1]})$.

For our method, we use Algorithm 1 and $s(x, \check{x}) = \frac{x+\check{x}}{2}$ to generate CF representation. We use the generated representation for training and inference as stated in Section 3.

For each method, we split the dataset randomly (with 80%-20% ratio) into a training dataset $D = \{(x^{(i)}, a^{(i)}, y^{(i)}) | i = 1, \ldots, n\}$ and a test dataset $D_{test} = \{(x^{(i)}, a^{(i)}, y^{(i)}) | i = n+1, \ldots, n+m\}$. The metrics for each method are calculated in five independent runs using the test dataset, and the average and standard deviation are reported in the tables. We use the mean squared error (MSE) for the regression task and the classification accuracy (Acc) for the classification task to evaluate the performance of our method and the baselines. To assess counterfactual fairness, we use total effect (TE) measure defined as $\text{TE} = \frac{1}{m} \sum_{i=n+1}^{m} |\hat{y}^{(i)} - \check{y}^{(i)[1]}|$, where $\check{y}^{(i)[1]}$ is the output of the predictor for the counterfactual sample $(\check{x}^{(i)[1]}, \check{a}^{(i)[1]})$. We also calculate the total effect for each protected group. More specifically, $\text{TE}_a = \frac{1}{|\{i|i>n,a^{(i)}=a\}|} \sum_{i>n,a^{(i)}=a} |\hat{y}^{(i)} - \check{y}^{(i)[1]}|$. In our experiments, $\text{TE}_0$ is the total effect for females, and $\text{TE}_1$ is the total effect for males. In general, smaller TE, $\text{TE}_0$, $\text{TE}_1$ imply that the method is fairer regarding the CF definition. Note that because CE method proposed in [26] makes predictions solely using non-descendants of $A$, empirical values for TE, $\text{TE}_0$, and $\text{TE}_1$ are 0 for the CE method.

**Results.** Table 1 and Table 2 represent the results for the UCI Adult Income dataset under CVAE and DCEVAE causal model, respectively. Note that there is no theoretical guarantee that UF, CA, ICA, and CR methods can satisfy Counterfactual fairness. As we can see in these two tables, the TE metric is significantly large for UF, CA, ICA, and they are not fair. On the other hand, CE, CR, and our methods can achieve relatively small TE values. However, among these three methods, our method can achieve the highest accuracy. Moreover, compared with the CR method, our method achieves roughly 80% improvement in terms of TE metric. It is worth mentioning that when we compare our algorithm with the UF model (which achieves the highest accuracy), our method only results in roughly 2% drop in accuracy but improves the TE metric by 92%. The results for the Law

Table 1: Logistic regression classifier results on UCI Adult dataset with DCEVAE causal model

| Method | Accuracy | TE | $TE_1$ | $TE_2$ |
|---|---|---|---|---|
| UF | **0.8165 ± 0.0037** | 0.2286 ± 0.0037 | 0.1903 ± 0.0061 | 0.2475 ± 0.0040 |
| CA | 0.8053 ± 0.0040 | 0.1847 ± 0.0075 | 0.1463 ± 0.0077 | 0.2037 ± 0.0090 |
| ICA | 0.8141 ± 0.0039 | 0.2085 ± 0.0035 | 0.1737 ± 0.0062 | 0.2256 ± 0.0025 |
| CE | 0.7858 ± 0.0028 | – | – | – |
| CR | 0.7914 ± 0.0015 | 0.0821 ± 0.0022 | 0.0506 ± 0.0034 | 0.0977 ± 0.0022 |
| Ours | 0.7931 ± 0.0030 | **0.0163 ± 0.0013** | **0.0146 ± 0.0021** | **0.0172 ± 0.0022** |

Table 2: Logistic regression classifier results on UCI Adult dataset with CVAE causal model

| Method | Accuracy | TE | $TE_1$ | $TE_2$ |
|---|---|---|---|---|
| UF | **0.8136 ± 0.0012** | 0.2338 ± 0.0044 | 0.1974 ± 0.0062 | 0.2517 ± 0.0069 |
| CA | 0.7946 ± 0.0021 | 0.1717 ± 0.0038 | 0.1203 ± 0.0069 | 0.1970 ± 0.0030 |
| ICA | 0.8095 ± 0.0026 | 0.2021 ± 0.0063 | 0.1540 ± 0.0082 | 0.2267 ± 0.0062 |
| CE | 0.7596 ± 0.0030 | – | – | – |
| CR | 0.7882 ± 0.0020 | 0.0868 ± 0.0041 | 0.0565 ± 0.0036 | 0.1017 ± 0.0048 |
| Ours | 0.7951 ± 0.0020 | **0.0126 ± 0.0042** | **0.0107 ± 0.0027** | **0.0139 ± 0.0060** |

School Success dataset under CVAE and DCEVAE are presented in Tables 3 and 4. Based on Table 3, only our algorithm and the CE method are able to achieve TE less 0.1. However, the MSE under our algorithm is significantly smaller. We also notice that under DCEVAE model and the law school success dataset (Table 4), the CR method exhibits similar performance as our method. It is worth mentioning that $TE_1$ and $TE_2$ under our algorithm are almost the same under our algorithm in all four tables. This shows our algorithm treats male and female groups almost the same in terms of CF.

We visualize the probability density function (pdf) of predicted FYA (output of the linear regression model) under the DECVAE causal model for factual (blue curve) and counterfactual (red curve) samples in Figure 5.[6] Under CF, we expect these two distributions to be the same. These figures show that our method is able to keep the model's behavior the same for factual and counterfactual samples.

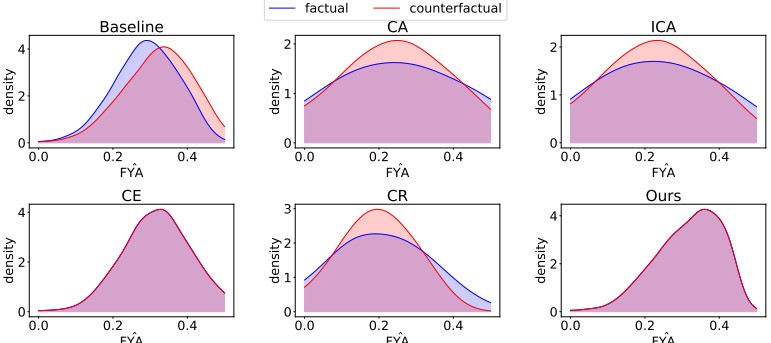

Figure 5: Density distribution of $\hat{FYA}$ with DCEVAE causal model

**Path-dependent Counterfactual Fairness.** Table 8 and Table 9 show the results for the UCI Adult Income dataset under path-dependent counterfactual. In this experiment, for finding counterfactual samples, we first consider a causal graph obtained after intervention on the variables that are not present on any unfair paths. Then, we use the resulting graph to generate counterfactual samples. In Tables 8 and 9, Acc (W) and TE (W) represent the accuracy and the total effect for a scenario in which Workclass is not on any unfair paths. Moreover, Acc (E) and TE (E) represent another scenario where Education is not on unfair paths. These tables show that, except for our algorithm and the CE method, other baselines fall short in satisfying the PCF. Our method exhibits significantly higher accuracy compared to the CE method as well. Similar results Additional results for Algorithm 2 are in the appendix. **Societal Impact:** In this paper, we introduced an algorithm that can find a

---

[5]The details about the network structure used for VAE can be found in the appendix.

[6]Figure 9 in the appendix illustrates the pdf of predicted FYA under the CVAE model.

Table 3: Linear regression results on Law School Success dataset with CVAE causal model

| Method | MSE | TE | TE$_1$ | TE$_2$ |
|--------|-----|-----|--------|--------|
| UF | **0.8664 ± 0.0040** | 0.1346 ± 0.0036 | 0.1345 ± 0.0028 | 0.1347 ± 0.0057 |
| CA | 0.8893 ± 0.0096 | 0.2335 ± 0.0125 | 0.2301 ± 0.0092 | 0.2362 ± 0.0016 |
| ICA | 0.8679 ± 0.0064 | 0.1608 ± 0.0058 | 0.1594 ± 0.0046 | 0.1619 ± 0.0078 |
| CE | 0.8900 ± 0.0054 | – | – | – |
| CR | 0.8622 ± 0.0148 | 0.1035 ± 0.0027 | 0.1032 ± 0.0024 | 0.1038 ± 0.0039 |
| Ours | 0.8692 ± 0.0060 | **0.0661 ± 0.0019** | **0.0654 ± 0.0023** | **0.0667 ± 0.0016** |

Table 4: Linear regression Results on Law School Success dataset with DCEVAE causal model

| Method | MSE | TE | TE$_1$ | TE$_2$ |
|--------|-----|-----|--------|--------|
| UF | **0.8677 ± 0.0043** | 0.1344 ± 0.0056 | 0.1345 ± 0.0064 | 0.1343 ± 0.0054 |
| CA | 0.8783 ± 0.0084 | 0.1759 ± 0.0368 | 0.1767 ± 0.0384 | 0.1753 ± 0.0357 |
| ICA | 0.8693 ± 0.0045 | 0.1459 ± 0.0169 | 0.1466 ± 0.0182 | 0.1454 ± 0.0161 |
| CE | 0.8781± 0.0068 | – | – | – |
| CR | 0.8703 ± 0.0055 | **0.0989 ± 0.0058** | **0.1000 ± 0.0070** | **0.098 ± 0.0052** |
| Ours | 0.8687 ± 0.0045 | **0.1076 ± 0.0039** | **0.1086 ± 0.0049** | **0.1068 ± 0.0032** |

fair predictor under counterfactual fairness. However, we do not claim CF is the only right fairness notion. Depending on the application, counterfactual fairness may or may not be the right choice for counterfactual fairness. **Limitation:** As stated in [26], to find a predictor under CF, causal model $\mathcal{M}$ should be known to the fair learning algorithm. Finding a causal model is challenging since there can be several causal models consistent with observational data [31]. This limitation exists for our algorithm and baselines.

# 6 Conclusion

We proposed a novel method to train a predictor under counterfactual fairness. Unlike [26], which shows that a sufficient condition for satisfying CF is to not use the features that are descendants of the sensitive attribute, our algorithm uses all the available features leading to better performance. The proposed algorithm generates a representation for training that guarantees CF and improves performance compared to the baselines. We also showed that our algorithm can be extended to path-dependent counterfactual fairness.

Table 5: Logistic regression classifier on UCI Adult dataset with CVAE causal model

| Method | Acc (W) | TE (W) | Acc (E) | TE (E) |
|--------|---------|--------|---------|--------|
| UF | **0.8136 ± 0.0012** | 0.2340 ± 0.0043 | **0.8136 ± 0.0012** | 0.1884 ± 0.0048 |
| CA | 0.7945 ± 0.0026 | 0.1715 ± 0.0046 | 0.7947 ± 0.0020 | 0.1600 ± 0.0057 |
| ICA | 0.8063 ± 0.0022 | 0.1997 ± 0.0107 | 0.8076 ± 0.0032 | 0.1689 ± 0.0116 |
| CE | 0.7596 ± 0.0030 | – | 0.7596 ± 0.0030 | |
| CR | 0.7882 ± 0.0020 | 0.0869 ± 0.0040 | 0.7926 ± 0.0021 | 0.1097 ± 0.0023 |
| Ours | 0.7951 ± 0.0030 | **0.0127 ± 0.0043** | 0.7952 ± 0.0022 | **0.0131 ± 0.0053** |

Table 6: Logistic regression classifier on UCI Adult dataset with DCEVAE causal model

| Method | Acc (W) | TE (W) | Acc (E) | TE (E) |
|--------|---------|--------|---------|--------|
| UF | **0.8165 ± 0.0037** | 0.2285 ± 0.0037 | **0.8165 ± 0.0037** | 0.2065 ± 0.0036 |
| CA | 0.8063 ± 0.0030 | 0.1871 ± 0.0030 | 0.8039 ± 0.0028 | 0.1570 ± 0.0096 |
| ICA | 0.8141 ± 0.0039 | 0.2087 ± 0.0041 | 0.8143 ± 0.0035 | 0.1766 ± 0.0042 |
| CE | 0.7858 ± 0.0028 | – | 0.7858 ± 0.0028 | – |
| CR | 0.7911 ± 0.0020 | 0.0824 ± 0.0028 | 0.7968 ± 0.0027 | 0.1026 ± 0.0021 |
| Ours | 0.7932 ± 0.0030 | **0.0164 ± 0.0014** | 0.7939 ± 0.0038 | **0.0159 ± 0.0016** |

## Acknowledgements

This work was funded in part by the National Science Foundation under award IIS-2202699, IIS-2301599, and ECCS-2301601, by a grant from the Ohio State University Translational Data Analytics Institute, by the National Center For Advancing Translational Sciences under award UL1TR002733.

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

# A    Related Work

This section succinctly presents an overview of fairness in the context of machine learning, especially focusing on the burgeoning concept of counterfactual fairness.

**Fairness in Machine Learning.** While there have been several efforts on developing fair machine learning algorithms (see, e.g., [23, 44, 42, 22, 3, 34]), fairness still remains an elusive and loosely defined concept. Several definitions have been proposed, each motivated by unique considerations and emphasizing different elements. For instance, one such notion is 'unawareness', which necessitates the exclusion of sensitive attributes from the input data fed into machine learning models. Conversely, parity-based fairness sets guidelines on how models should perform across different demographics. Demographic parity, a widely accepted group fairness criterion, ensures consistent distribution of predictions ($\hat{Y}$), regardless of sensitive attributes ($A$), defined as $P(\hat{Y}|A = 0) = P(\hat{Y}|A = 1)$ [13].

Equal opportunity is another key criterion, ensuring that individuals from diverse protected attribute groups have the same probability of selection, represented by $P(\hat{Y} = 1|A = 0, Y = 1) = P(\hat{Y} = 1|A = 1, Y = 1)$ [20]. The related concept of equal odds prescribes equal true positive and false positive rates across different protected attribute groups [38].

Additionally, preference-based fairness argues that an algorithm's design should not be solely determined by its creators or regulators but should also incorporate the preferences of those directly affected by the algorithm's outputs [43, 12].

**Counterfactual Fairness.** Emerging from the broader concept of individual fairness, counterfactual fairness seeks to guarantee that similar datapoints are treated identically [32]. It uses counterfactual pairs to establish similarity. The field of natural language processing has seen the use of intuitive causal models, where words associated with protected attributes are substituted to generate counterfactual data [29, 14]. However, this approach might overlook possible causal relationships among words.

To overcome this, a more sophisticated causal model was proposed by [31], facilitating a three-step process for counterfactual data generation: Abduction-Action-Prediction. This process includes the inference of exogenous variables' distribution based on observed data, the modification of protected attributes, and the computation of resultant attributes. Leveraging this causal model, [26] proposed the formal definition of counterfactual fairness.

Adopting the advanced assumptions of [31] about structural functions in the causal model allows for counterfactual data generation via the Markov chain Monte Carlo (MCMC) algorithm [15]. Several practical applications employ an encoder-decoder-like structure for counterfactual inference, with the encoder's hidden representation considered as the exogenous variable [30, 21, 37, 24]. The decoder predicts counterfactual data after the sensitive attribute is modified. Certain research, such as [8], explores counterfactual inference without a causal model by reconceptualizing it as a multi-objective issue.

A myriad of techniques exist to construct fair models using counterfactual inference. The simplest among these, unawareness, involves the removal of the protected attribute from the input [18]. Yet, due to potential correlations between remaining features and protected attributes, this method often falls short. [26] suggested an approach that uses only the non-descendant variables of the sensitive attribute as model inputs, achieving perfect fairness. Other research aimed to attain approximate counterfactual fairness [36].

A common approach employed by [14, 36, 6, 11, 33] introduces a fairness penalty regularizer to the loss function when training a counterfactually fair model. Meanwhile, studies by [24, 14, 35, 10] have used a counterfactual augmentation method to increase fairness. This involves generating a new training dataset by mixing counterfactual and factual data. Notably, several studies [17, 39, 41, 19, 28] sought to minimize the correlation between exogenous variables and the sensitive attribute using adversarial learning or regularization. They propose another kind of fairness from the causal perspective, differing from [26].

While the method proposed by [26] maintains perfect fairness, it often leads to a significant decrease in precision due to the underutilization of observed data's information. Subsequent methodologies have managed to enhance precision, though they cannot theoretically guarantee counterfactual fairness. As

counterfactual fairness gains increasing traction [17, 6, 27], there is an urgent need for approaches that simultaneously augment performance and uphold fairness.

## B  Proofs

*Theorem 1.* We start by finding $\Pr\{R_{A \leftarrow a'}(U) = r | X = x, A = a\}$.

$$\Pr(R_{A \leftarrow a'}(U) = r | X = x, A = a) = \int \Pr(R_{A \leftarrow a'}(u) = r | X = x, A = a) \Pr(U = u | X = x, A = a)$$

Note that $R$ in Algorithm 1 depends on value realization $u$ and feature vector $x$. However, given $u$, $R_{A \leftarrow a'}(u)$ does not depend on intervention $A = a'$ as $s(.)$ is a symmetric function and gets all the counterfactual samples for different values of $a'$. As a result, $\Pr(R_{A \leftarrow a'}(U) = r | X = x, A = a)$ does not depend on $a'$. Moreover, $\Pr(U = u | X = x, A = a)$ is not a function of $a'$ and does not depend on $a'$. This implies the right-hand side of (3) does not change by $a'$. As a result, for a given pair of $(x, a)$, $\Pr(R_{A \leftarrow a'}(U) = r | X = x, A = a)$ remains unchanged for all $a' \in \mathcal{A}$. This implies that $R$ satisfies CF. Consequently, any function of $R$ including $g_w(R)$ satisfies CF.  □

*Theorem 2.* Assume that $R$ has been generated using Algorithm 2. We have,

$$\Pr(R_{A \leftarrow a', X_{\mathcal{P}_{\mathcal{G}_A}^c} \leftarrow x_{\mathcal{P}_{\mathcal{G}_A}^c}}(U) = r | X = x, A = a) =$$

$$\int \Pr(R_{A \leftarrow a', X_{\mathcal{P}_{\mathcal{G}_A}^c} \leftarrow x_{\mathcal{P}_{\mathcal{G}_A}^c}}(u) = r | X = x, A = a) \Pr(U = u | X = x, A = a)$$

Note that, $\Pr(R_{A \leftarrow a', X_{\mathcal{P}_{\mathcal{G}_A}^c} \leftarrow x_{\mathcal{P}_{\mathcal{G}_A}^c}}(U) = r | X = x, A = a)$ does not depend on $a'$. This is because, Algorithm 2 generates a presentation using a symmetric function, and all the counterfactual samples $\{(\check{x}_{\mathcal{P}_{\mathcal{G}_A}}^{[1]}, \check{a}^{[1]}), \ldots, (\check{x}_{\mathcal{P}_{\mathcal{G}_A}}^{[|\mathcal{A}|-1]}, \check{a}^{[|\mathcal{A}|-1]})\})$, and intervention on $A$ does not change the presentation. Note that $\Pr(U = u | X = x, A = a)$ is not a function of $a'$ and does not depend on $a'$. This implies the right-hand side of (3) does not change by $a'$. As a result, for a given pair of $(x, a)$, $\Pr(R_{A \leftarrow a', X_{\mathcal{P}_{\mathcal{G}_A}^c} \leftarrow x_{\mathcal{P}_{\mathcal{G}_A}^c}}(U) = r | X = x, A = a)$ remains unchanged for all $a' \in \mathcal{A}$. This implies that $R$ satisfies PCF. Consequently, any function of $R$ including $g_w(R)$ satisfies PCF.

$$\Pr\{\hat{Y}_{A \leftarrow a, X_{\mathcal{P}_{\mathcal{G}_A}^c} \leftarrow x_{\mathcal{P}_{\mathcal{G}_A}^c}}(U) = y | X = x, A = a\} = \Pr\{\hat{Y}_{A \leftarrow a', X_{\mathcal{P}_{\mathcal{G}_A}^c} \leftarrow x_{\mathcal{P}_{\mathcal{G}_A}^c}}(U) = y | X = x, A = a\}$$

□

## C  Synthetic Data Simulation

For the real data experiments in Section 5, the causal model behind the problem remains unknown, and we had to make an assumption about the causal structure and estimate the causal model parameters using observed data.

In order to make sure that we are working with a true causal model and true structural equations and demonstrate our proposed method can improve performance while maintaining counterfactual fairness, we carry out a simulation experiment on the synthetic data.

We consider a causal graph shown in Figure 6. The structural function defined in the corresponding causal model is

$$f_X = \sin U_1 + \cos U_2 A + A + 0.1; f_Y = 0.2X^2 + 1.2X + 0.2$$

To generate the synthetic dataset, we sampled $A$ from the Bernoulli distribution with $p = 0.4$ for 3000 times. $U_1$ and $U_2$ are sampled independently from the normal distribution $\mathcal{N}(0, 1)$. $X$ and $Y$ were computed with the structural function. The counterfactual data $\check{X}$ were computed by substituting $A$ in the structural function with $\check{A}$.

We implemented our method and the baseline methods as described in Section 5 (since there is no difference between observed data and factual data in this scenario, we have no ICA baseline here). For the CR method, we set the weight of the fairness regularization term as 0.05. The 3000 synthetic data were split into a training set and a test set with a ratio of 80% - 20%. Then we trained a linear

Table 7: Linear regression results on the synthetic datal

| Method | MSE (L) | TE | $TE_0$ | $TE_1$ |
|--------|---------|-----|--------|--------|
| UF | $0.0172 \pm 0.0009$ | $1.2499 \pm 0.0093$ | $1.2460 \pm 0.0252$ | $1.2543 \pm 0.0310$ |
| CA | $0.3467 \pm 0.0208$ | $0.5372 \pm 0.0057$ | $0.5409 \pm 0.0125$ | $0.5316 \pm 0.0160$ |
| CE | $0.7868 \pm 0.0556$ | $0.000 \pm 0.0000$ | $0.000 \pm 0.0000$ | $0.000 \pm 0.0000$ |
| CR | $0.8598 \pm 0.0521$ | $0.2572 \pm 0.0036$ | $0.2590 \pm 0.0066$ | $0.2544 \pm 0.0078$ |
| Ours | $0.4739 \pm 0.0202$ | $0.000 \pm 0.0000$ | $0.000 \pm 0.0000$ | $0.000 \pm 0.0000$ |

regression model with the training data and calculate MSE, TE, $TE_0$, $TE_1$ with the test data. For each method, we run the experiments five times with different random splits. Table 7 provided the results for the simulation. With the ground truth of the causal model, our proposed method could achieve a 100% counterfactual fairness as the CE method. However, with the use of $X$, we improved the MSE to a large extent, almost as well as the CA method.

# D  Detailed Experimental Setup on Real Data

We conducted our experiments using a supercomputing platform. The CPUs used were Intel(R) Xeon(R) Platinum 8268 CPU @ 2.90GHz, and the GPU model was a Tesla V100. Our primary software environments were Python 3.9, Pytorch 1.12.1, and CUDA 10.2.

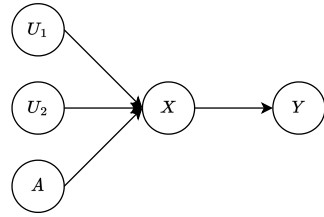

The VAE structure for the CVAE model is shown in Figure 7. The details for training the VAE can be found in [37]. However, we briefly discuss how the training was done. An encoder takes $[X_\alpha, X_\beta, Y, A]$ as input to generate the hidden variable $U$. The decoders then serve as structural functions. Decoder $f_\alpha$ takes $U$ as

Figure 6: Synthetic causal graph

input to generate $\check{X}_\alpha$, decoder $f_\beta$ takes $[U, \check{A}]$ to generate $\check{X}_\beta$, and $f_Y$ also uses $[U, \check{A}]$ to generate $\check{Y}$.

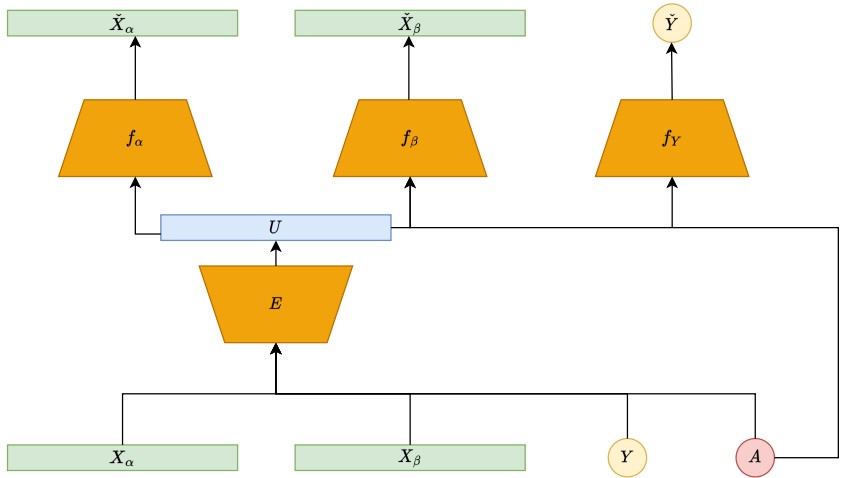

Figure 7: VAE Structure for CVAE Model

During the training of the VAE, we used the following loss function:

$$L = w_\alpha l_\alpha(X_\alpha, \check{X}\alpha) + w_\beta l_\beta(X_\beta, \check{X}\beta) + w_Y l_Y(Y, \check{Y}) + w_u \text{KL}(U||U_p) + W_{fair}||\check{Y}^{[0]} - \check{Y}^{[1]}||_2$$

For the Law School Success dataset, $l_\alpha$ is the BCE loss function, and $l_\beta$ and $l_Y$ are the MSE loss function. For the UCI Adult Income dataset, $l_\alpha$, $l_\beta$, and $l_Y$ are BCE loss functions. We set $w_\alpha = 1, w_\beta = 1, w_Y = 1, w_u = 1$, and $w_{fair} = 0.15$. The batch size was set to 256 and the learning rate to 0.001. The experiments for the UF, CA, ICA, and CR methods were based on the same VAE.

For the CE and our method, as we needed to use the VAE during the test time, we removed the use of $Y$ from the structure, including the decoder $f_Y$. Hence, the encoder uses $[X_\alpha, X_\beta, A]$ to obtain $U$, and $\check{X}_\alpha = f_\alpha(U)$, $\check{X}_\beta = f_\beta(U, \check{A})$. In this case, the loss function becomes:

$$L = w_\alpha l_\alpha(X_\alpha, \check{X}\alpha) + w_\beta l_\beta(X_\beta, \check{X}_\beta) + w_u \text{KL}(U||U_p)$$

For the Law School dataset, we kept the hyperparameters the same, so $w_\alpha = 1, w_\beta = 1, w_u = 1$. For the UCI Adult Income dataset, we set $w_\alpha$ to 1, $w_\beta$ to 1, and $w_u$ to 1.

Figure 8 depicts the VAE structure for the DCEVAE model. The details for training the VAE can be found in [24], and we summarize it here. The hidden variable is divided into two parts, $U_\alpha$ and $U_\beta$. Hence, $U_\alpha = E_\alpha(X_\alpha, Y)$ and $U_\beta = E_\beta(X_\beta, A, Y)$. During the decoding stage, $\check{X}_\alpha = f_\alpha(U_\alpha)$, $\check{X}_\beta = f_\beta(U_\beta, \check{A})$, and $\check{Y} = f_Y(U_\alpha, U_\beta, \check{A})$. A discriminator, $D_\psi$, is also employed to aid in disentangling $U_\alpha$ and $U_\beta$.

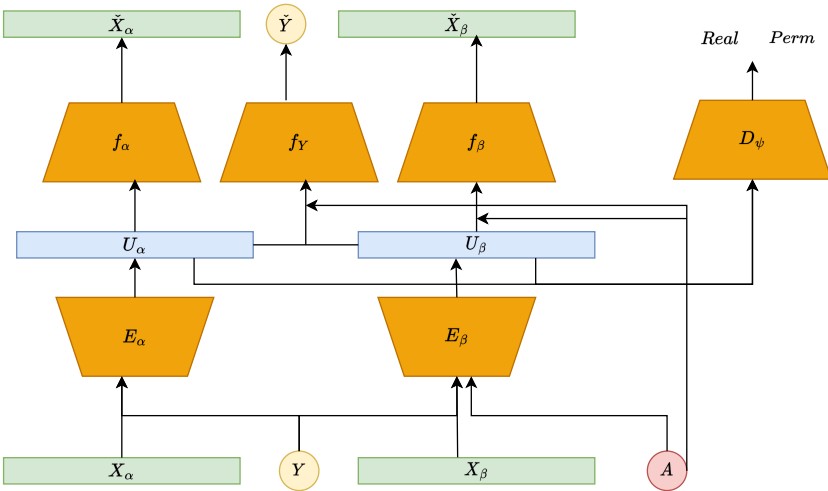

Figure 8: VAE Structure for DCEVAE Model

The training of this VAE can be divided into two stages. In the first stage, we permuted the $U_\beta$ generated in the batch of data and concatenated them with $U_\alpha$. The discriminator was trained to distinguish whether a $[U_\alpha, U_\beta]$ is randomly permuted. In the second stage, we trained the encoders and decoders. The loss function is:

$$\begin{aligned} L &= w_\alpha l_\alpha(X_\alpha, \check{X}_\alpha) + w_\beta l_\beta(X_\beta, \check{X}_\beta) + w_Y l_\alpha(Y, \check{Y}) + w_u \text{KL}(U||U_p) \\ &+ W_{fair}||\check{Y}^{[0]} - \check{Y}^{[1]}||_2 + w_h * \text{TC} \end{aligned}$$

Here, TC refers to the total correlation loss, which is the negative discrimination loss of $D_\psi$[7]. We used the same $l_\alpha, l_\beta, l_Y$, and weights $(w_\alpha, w_\beta, w_Y)$ as those used for training the CVAE. We also used the same batch size and learning rate. $w_h$ is set at 0.4, and $w_{fair}$ is set at 0.2. As before, we used the same VAE for the implementation of UF, CA, ICA, and CR methods.

For the CE and our method, we removed all structures related to $Y$, as we did with the CVAE. For the Law School Success dataset, we kept the hyperparameters the same. And for the UCI Adult Income dataset, we set $w_u$ to 0.5 and $w_h$ to 0.4.

For the finding predictors, we used the linear regression model for the Law School Success Dataset and the logistic regression model for the UCI Adult Income dataset. When training the CR model, we set the coefficient of the regularization term as 0.002.

We split each dataset into a training set, validation set, and test set with a ratio of 60%-20%-20%. The validation set was used to stop the training of the VAE early. The training and validation sets were used together to train the predictors. All experiments were repeated five times with different splits to ensure the results are stable.

---

[7]More details about the function can be found in [24]

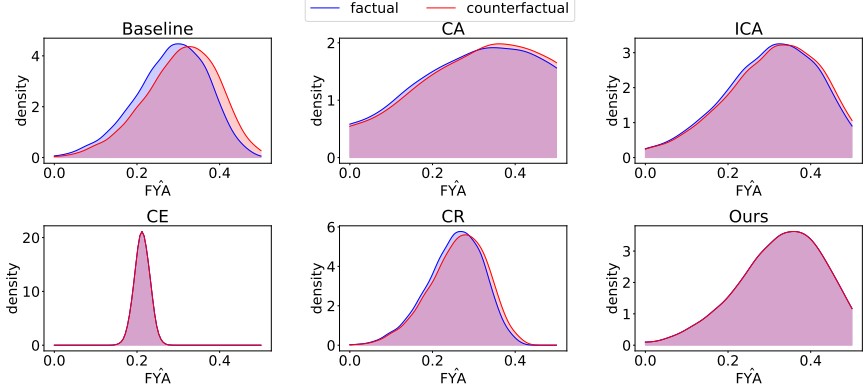

Figure 9: Density distribution of $\hat{FYA}$ with CVAE causal model

Figure 9 visualizes the PDF of the predicted FYA under the CVAE causal model. As seen in Figure 5, our model is more effective in maintaining the model's behavior for both factual and counterfactual data.

Table 8: Linear regression results on Law School Success dataset with CVAE causal model

| Method | MSE (G) | TE (G) | MSE (L) | TE (L) |
|--------|---------|--------|---------|--------|
| UF | **0.8664 ± 0.0060** | 0.1331 ± 0.0034 | **0.8664 ± 0.0060** | 0.1258 ± 0.0039 |
| CA | 0.8889 ± 0.0097 | 0.2330 ± 0.0126 | 0.8915 ± 0.0098 | 0.2358 ± 0.0127 |
| ICA | 0.8704 ± 0.0042 | 0.1633 ± 0.0014 | 0.8683 ± 0.0065 | 0.1543 ± 0.0044 |
| CE | 0.8900 ± 0.0076 | − | 0.8900 ± 0.0076 | − |
| CR | 0.8693 ± 0.0064 | 0.1035 ± 0.0027 | 0.8696 ± 0.0063 | 0.0880 ± 0.0025 |
| Ours | 0.8689 ± 0.0059 | **0.0663 ± 0.0019** | 0.8682 ± 0.0060 | **0.0655 ± 0.0019** |

Table 9: Linear regression results on Law School Success dataset with DCEVAE causal model

| Method | MSE (G) | TE (G) | MSE (L) | TE (L) |
|--------|---------|--------|---------|--------|
| UF | **0.8677 ± 0.0043** | 0.0780 ± 0.0086 | **0.8677 ± 0.0043** | 0.1300 ± 0.0053 |
| CA | 0.8748 ± 0.0050 | 0.1151 ± 0.00277 | 0.8794 ± 0.0010 | 0.1736 ± 0.0398 |
| ICA | 0.8687 ± 0.0046 | 0.0934 ± 0.0160 | 0.8696 ± 0.0047 | 0.1372 ± 0.0166 |
| CE | 0.8781 ± 0.0068 | − | 0.8781 ± 0.0068 | − |
| CR | 0.8708 ± 0.0042 | **0.0463 ± 0.0049** | 0.8712 ± 0.0053 | **0.0821 ± 0.0052** |
| Ours | 0.8679 ± 0.0045 | 0.0693 ± 0.0037 | 0.8692 ± 0.0047 | 0.0968 ± 0.0024 |

Tables 8 and 9 present the results for the Law School Success dataset under path-dependent counterfactuals. In these tables, MSE(L) and TE(L) represent the MSE and TE when the LSAT is not in any unfair path, while MSE(G) and TE(G) correspond to the scenario in which GPA is not in any unfair path. The results affirm that our method consistently satisfies PCF in every case. Although the CR method can achieve PCF similar to our method when GPA or LSAT is not in any unfair path of the DCEVAE causal model, it fails in other scenarios because it does not guarantee counterfactual fairness.

