# OpenReview forum: "Counterfactually Fair Representation"
_NeurIPS.cc/2023/Conference — NeurIPS 2023 poster_

### Official Review · Reviewer_jctR · 2023-07-03

**Soundness:** 3 good
**Presentation:** 3 good
**Contribution:** 3 good
**Rating:** 6
**Confidence:** 2

**Summary:**

The paper presents a new approach for achieving counterfactual fairness.  Specifically, builds on past work on counterfactual fairness and proposes an improved algorithm that does not discard any input features. The paper achieves this by sampling from the latent background variables to aggregate a set of counterfactual features. The features are then passed through a symmetric function and concatenated with the latent factor to form a data representation. The representations are then used to train a model that is guaranteed to achieve counterfactual fairness. The paper also presents the notion of path-dependent counterfactual fairness and extends their original algorithm to this fairness notion.

**Strengths:**

**Strengths**:

1. The paper is fairly easy to follow and provides detailed background on counterfactual fairness.
2. The proposed path-dependent counterfactual fairness is a welcome extension of counterfactual fairness that does not neglect features for being dependent on the protected attribute.
3. The paper showcases extensive experiments on multiple setups to show the efficacy of the proposed approach.

**Weaknesses:**

**Weakness**:

1. The motivation behind using a symmetric function needs to be provided in more detail. Why is it better to form a single representation than to augment the individual counterfactual samples in the training set? Also, if the proposed approach takes the mean of all features, isn’t it losing a lot of information? Will this work well when most (or all) of the input features are dependent on the protected attribute A? Are there any synthetic experiments to show evidence?
2. It is unclear why the paper does not report TE results for CE in any of the results tables. It would be interesting to see the trade-off between fairness and utility with this baseline, which ignores all the descendants of A.
3. It would interesting to have a qualitative discussion on the impact of path-dependent counterfactual fairness. Also, how the performance of the proposed PCF and CF algorithms differs from each other?
4. The presentation towards the beginning of the paper can be smoothened out for readers not too familiar with CF literature. For example, in line 86: it is unclear how Y and Z fit into the graphical model. Are they subsets of the unobserved U or observed variables V? Alternatively, the paper can also state a real-world example in the beginning (section 2 instead of section 3 as it is now) and use parts of it to guide the reader through the rest of the paper.
5. The presentation of the proofs can be formalized a bit by having an explicit proof section for each theorem/lemma. The paper can write the proofs in blocks of (begin{proof} \end{proof}).

**Questions:**

Question:

1. Clarification question about the sampling of u. When u is sampled from the posterior distribution? Is it kept fixed while the dependent x features are generated for each of the labels of the protected attribute A=a’ or is a different u used for every sample?

Please reply to any questions in the weakness section as well.

---

> ### Author Rebuttal · Authors · 2023-08-09
>
> Thank you for your comment. Here are our answers to your questions.
>
> Question 1: As we can see in algorithms 1 and 2, $u$ is generated based on conditional distribution U given X=x and A=a. As a result, $u$ is generated differently for each sample. As we discussed in the paper and the appendix, we can generate $u$ from the conditional distribution $\Pr(U|X=x, A=a)$ using VAEs.
>
> #### Weakness 1:
> **The motivation behind using a symmetric function needs to be provided in more detail.**  Our motivation for using such a symmetric function comes from the definition of counterfactual fairness (Definition 1 in the paper). Counterfactual fairness implies that the decision should be the same in the factual and counterfactual world. Algorithms 1 and 2 find a representation that is a symmetric function of the factual and counterfactual world. This means that if replace the counterfactual world with the factual world, the representation remains the same and does not change.
>
> **Why is it better to form a single representation than to augment the individual counterfactual samples in the training set?** Counterfactual factual data augmentation is one of our baselines. There is no theoretical guarantee that Counterfactual factual data augmentation can satisfy counterfactual fairness (as we can see in our experiment). However, our representation satisfies exact counterfactual fairness (proved in Theorem 1 and 2).
>
> **Also, if the proposed approach takes the mean of all features, isn’t it losing a lot of information?** Always there is a trade-off between fairness and accuracy. However, as you can see in the experiment, our method is able to satisfy CF and the obtained accuracy is very close to the accuracy of a model trained without any fairness constraint.
>
> **Will this work well when most (or all) of the input features are dependent on the protected attribute A? Are there any synthetic experiments to show evidence?** Yes, it still works well as the representation includes $u$ in addition to the features. Please check the appendix for experiments with synthetic data. In that experiment, $A$ directly impacts all the features.
>
> #### Weakness 2:
> We explained why we did not fill in the TE values for CE methods in the paper (line 286).  CE method always has 0 TE by the definition (Not by empirical calculation) since it does not use any observable feature. We will enter 0.0 for TE for CE in the tables in the camera-ready version.
>
> #### Weakness 3:
> Path-dependent counterfactual fairness has been proposed by [22] (https://arxiv.org/pdf/1703.06856.pdf). In the last paragraph of page 15 of [22], the authors provide an example from UC Berkeley’s admission process to show the importance of path-dependent CF. We will add this example to our paper and will discuss the difference between CF and path-dependent CF in more detail. Generally, based on UC Berkeley’s admission, if all the causal path from sensitive attributes to target $Y$ is not considered unfair, then we should consider using Path dependent CF.
>
> #### Weakness 4:
> As we mentioned in line 86, $Y$ and $Z$ here are observable variables and are subsets of $V$. Thank you for your advice. We will re-write this part, and add an example similar to Example 1 to section 2.1. Example 1 in the paper is coming from [22] which is based on a real-world scenario and can be used to explain observable and unobservable variables.
>
> #### Weakness 5:
> Thank you for your comment. Yes, we take this comment very seriously and will write the proof in a more organized way in the camera-ready version.

---

### Official Review · Reviewer_TTY3 · 2023-07-04

**Soundness:** 3 good
**Presentation:** 4 excellent
**Contribution:** 3 good
**Rating:** 7
**Confidence:** 3

**Summary:**

The paper introduces an innovative training technique that guarantees adherence to the counterfactual fairness measure. The method comprises three distinct steps: counterfactual samples generation, counterfactually fair representation generation, and fair model training.
In contrast to previous approaches that exclude descendants of the sensitive attribute under investigation, authors investigate the possibility of training a predictor that uses all attributes and simultaneously reaches counterfactual fairness. Indeed, the proposed approach leverages all attributes while achieving enhanced performance. Additionally, it is extended to incorporate the concept of path-dependent counterfactual fairness.
Theoretical and empirical evidence is provided to demonstrate the correctness and efficacy of the method.


**Strengths:**

The contribution is framed with the necessary background notions and critically positioned with respect to relevant previous literature.
Compared to existing solutions, the proposed solution overcomes the restriction of using only non-descendants to achieve counterfactual fairness.
Also, it introduces a path-dependent variation of the measure, offering a valuable and novel, original contribution to the field. The authors further demonstrate the theoretical and empirical efficacy of the proposed method.
Through the experiments, the authors compare the results of their approach with those of competitors, thereby subjecting the method's outcomes to rigorous testing against relevant techniques. The implementation choices, such as those related to the type of causal models selected, are well justified. The description and interpretation of the conducted results are comprehensive and presented clearly and straightforwardly.
The paper's explicit problem definition is essential to understand the proposed method and its algorithms. Lastly, the examples reported multiple times during the method presentation provide a clear and coherent framework for the reader to follow and fully comprehend the proposed approach.

**Weaknesses:**

In presenting the method, it is often challenging to differentiate between the innovative aspects proposed and the background knowledge derived from existing literature. The authors could more explicitly highlight the original contributions.

**Questions:**

(1) The value of MSE needs to be properly contextualized, as its interpretation depends on the range. The authors should briefly describe the dataset properties and report the results achieved by state-of-the-art solutions to facilitate interpretation.
(2) The chosen baseline for ADULT is weak and needs better motivation and contextualization. The baseline's performance levels are lower than the state-of-the-art results. In this context, the Fairness scores do not differ significantly and do not seem to be indicative of how well the method works or representative of the different impacts on performance. Perhaps by using a more complex model or a stronger baseline, the impacts on Accuracy would be more evident and further demonstrate the contribution's novelty.

**Limitations:**

Although the authors have included considerations on the limitations and societal impact of the proposed work, the discussion appears too concise and could benefit from further elaboration.
For example, regarding the reported limitation of adopting a causal model, it is a general drawback of causality that, although valid and important to mention, may not provide sufficient information or relevance to the specific contextualized method presented in the paper.
Furthermore, the authors indicate that this limitation applies to both their method and the baselines. However, not all baselines rely on the same assumptions as the proposed method; for example, not all baselines require structural equations. Therefore, this limitation is not equally suffered by all techniques.
Finally, the assumption of knowing the causal model, understood in terms of structural equations, is a condition that is practically impossible in real-world contexts. Since real-world applications drive the motivation for pursuing fairness, there remains a significant gap that needs to be more adequately stated despite the authors mentioning this limitation.

---

> ### Author Rebuttal · Authors · 2023-08-09
>
> Thank you for your comment. Here are our answers to your questions.
>
> Question 1: $Y$ in the law school dataset represents the (normalized) first-year average GPA (FYA). The range of it in the dataset is from –3.45 to 3.48 (We will add more information about the dataset to include the range of $Y$ and other features.).
> The UF baseline is the best MSE that we can get using a linear regression model since it does not take into account any fairness constraint. Please note, the goal of our experiment is to measure the fairness-accuracy trade-off. Accuracy alone is not informative. CE, CA, ICA, and CR baselines are the recent works (or the sate-off the art) that try to satisfy counterfactual fairness while improving the fairness-accuracy trade-off. Therefore, as long as we train the same model (e.g., linear regression) with the same training dataset using the baseline methods, we can correctly compare the fairness-accuracy trade-off. Yes, we maybe can improve the accuracy using a more complex model. However, still we need to look at the trade-off between accuracy and fairness.
>
> Question 2:  We want to again emphasize that the goal of our experiment is to measure the fairness-accuracy trade-off for different methods (looking at accuracy without measuring fairness is not appropriate). Therefore, we measured the accuracy and TE of baselines and our method under the same situation. We trained a linear model using our algorithm and baseline algorithms. We kept the training data the same for all the methods. The training dataset was pre-processed in the same manner as that in this paper (https://arxiv.org/abs/2011.11878). Only under the same situation, the baselines and our method can be compared.
>
> That is true that we can improve accuracy using a non-linear model. But improving accuracy using a more complex model is not the goal of the experiment. We are mainly concerned with the fairness-accuracy trade-off. For the sake of completeness, we will add the following results which are for neural networks:
>
> | Method | Accuracy            | TE                  | TE0                 | TE1                 |
> |--------|---------------------|---------------------|---------------------|---------------------|
> | UF     | $0.8199 \pm 0.0013$ | $0.1494 \pm 0.0103$ | $0.1187 \pm 0.014$  | $0.1644 \pm 0.0103$ |
> | CA     | $0.8142 \pm 0.0027$ | $0.1838 \pm 0.0103$ | $0.1312 \pm 0.0050$ | $0.2098 \pm 0.0151$ |
> | CE     | $0.7933 \pm 0.0010$ | 0                   | 0                   | 0                   |
> | Ours   | $0.7955 \pm 0.0017$ | $0.0256 \pm 0.0031$ | $0.0212 \pm 0.0043$ | $0.0278 \pm 0.0037$ |
>
> Regarding limitation: All the baselines except UF (which trains a model without any fairness constraint) rely on an underlying causal model for generating counterfactual examples. We will clarify this in our paper. Also, we will further emphasize that if the causal model is not known, human knowledge and causal structure learning methods (see e.g., https://arxiv.org/pdf/1706.09141.pdf) can help to find the causal structure.

---

### Official Review · Reviewer_rDmX · 2023-07-08

**Soundness:** 3 good
**Presentation:** 4 excellent
**Contribution:** 3 good
**Rating:** 5
**Confidence:** 3

**Summary:**

The paper proposes a new algorithm to learn counterfactual fairness representation. The idea behind this is first to generate the counterfactual examples and use the symmetric function to generate the symmetric function. Empirical results models trained with this method can better satisfy counterfactual fairness compared to the other methods.

**Strengths:**

* The proposed method is novel since it is the first work to focus on learning counterfactual fair representation, while the existing works focus on data augmentation or regularization. It also advances state-of-the-art results in terms of counterfactual unfairness mitigation.

* Compared to the existing methods, the proposed method is technically sound since the claims are supported by both theoretical analysis and experimental results.

* The submission is clearly written and well organized.


**Weaknesses:**

* Overall, the methods seem flexible to tune the utility and fairness trade-offs. Although the methods achieve the best performance in terms of effect total effect measure, the method lacks a tunable parameter to tune the utility and fairness trade-offs. In comparison, I believe counterfactual regularizer does not share the problem.

* The author claims their theoretical analysis as the guarantee to ensure counterfactual fairness. However, I think the word *guarantee* might be too strong. The reason is that in the experiment, the method does not ensure perfect counterfactual fairness (e.g., TE=0). The authors fail to provide such an explanation why there is a gap between the theory and experiments (e.g., where does unfairness in their method come from).

* I also think the result presentation could be better. Only listing the number in the table is not the best way to present the results. For example, the author can add the percentage of improvement of all metrics compared to the UF baseline for better presentation. Using Figures to visualize the trade-offs between fairness and utility is also a good choice.


**Questions:**

* Please respond to the first point in the Weaknesses section (e.g., tunable utility-fairness trade-off parameter).

* Please respond to the second point in the Weaknesses section (e.g., the gap between the experiment and theory).


**Limitations:**

The authors adequately addressed the limitations.

---

> ### Author Rebuttal · Authors · 2023-08-09
>
> Thank you for your comment. Here are our responses to your questions.
>
> Question 1: Our goal in our paper is to have a perfect fair algorithm, so we did not include tunable parameters to adjust between accuracy and fairness like the weights used in the regularization-like method (Note that using a regularizer cannot satisfy perfect fairness). However, our idea is easy to be extended to sacrifice some fairness to achieve better accuracy. In particular, we can slightly break the symmetry of the function $s$. For example, with the same setting of the experiment in the Appendix, we can replace the $0.5x + 0.5\check{x}$ with $0.6x + 0.4\check{x}$, which can sacrifice TE from 0 to 0.26 while improving the MSE from 0.46 to 0.31.
>
> Question 2: The reason why TE for our method is not 0 is that we used VAE to model the structural functions of the causal model. Since VAE can only approximate the underlying causal mechanisms, we cannot compute the exact counterfactual features, which results in the non-zero value of TE. In the appendix, we used another experiment on the synthetic data showing our algorithm can achieve perfect fairness when the exact causal model $\mathcal{M}$ is known.
>
> And thank you for your comment regarding the presentation. We will present the results in the format that you mentioned in the camera-ready version. We have plotted the mse-fairness trade-off curve for our method and the CR method and put it in the anonymous link (https://anonymous.4open.science/r/CF_Representation-B01B/trade_off.png ).

---

### Official Review · Reviewer_7vQZ · 2023-07-26

**Soundness:** 3 good
**Presentation:** 2 fair
**Contribution:** 2 fair
**Rating:** 5
**Confidence:** 4

**Summary:**

The authors claim to:
* present a novel and efficient method for generating counterfactually fair representations through the use of a symmetric function.
* theoretically show that ML models trained with such representations can achieve perfect/exact CF.
* extend their method to path-dependent counterfactual fairness as a bias mitigation strategy for unfair paths in a given causal graph.
* provide empirical validation with different causal models, datasets, and fairness notions to compare their method with other CF methods. They claim that 1) their method outperforms the method of only using non-descendants of sensitive attributes; 2) existing heuristic methods for training ML model under CF fall short of achieving perfect CF fairness.
The authors use mean squared error (for regression) and overall classification accuracy, and additionally total effect for each protected group, to compare their method against a collection of baseline methods.

**Strengths:**

The authors evaluate their method against a variety of baseline approaches, using two different causal models and two different datasets which provides a multiple points of reference for empirical comparison. Assuming the choice of representation is justified and the causal model distribution is obtainable, the approach presented in Algo 1 for generating a counterfactual representation to use within the unconstrained ERM problem is fairly straightforward. The authors provide acknowledgement of potential societal impacts, briefly noting that counterfactual fairness is not the only (or most appropriate) fairness notion depending on the task.

**Weaknesses:**

Abstract: The abstract is framed too closely around other works and does not effectively set the stage of this paper and its contributions. Note that only the last two sentences introduce aspects of this paper, while the remainder broadly introduce content that would be better suited for the introduction or related work section. I suggest beginning with the general problem setting, why it matters, implications of the current status quo (and thus why your work is relevant), and then outline the contributions to the problem/field that is presented in the paper.

Experiments: Please add justification for your choice of metrics - why are these meaningful or "good fits" at measuring CF between the different baselines? Note that even if these are the best choice for the included tasks, excluding the rationale/significance diminishes this.

It would be informative, interesting, and straightforward to also compute each sensitive attribute-specific accuracy in addition to the overall as another indicator of model bias for all methods.

Please consider using an additional and/or different dataset in the future. Consider referring to https://openreview.net/pdf?id=bYi_2708mKK for an argument on why the UCI Adult dataset may be worthwhile to "retire" in the context of fairness evaluation.

Conclusion: too short and brief, it reads as though the authors ran out of space.


- Provide rationale and citations for the use of symmetric functions in the context of representation learning. This is a nontrivial component of your proposed method, and without justification of this choice, it's challenging to make informed judgments around your method.

- The transition from Definition 1 and minimizing the provided empirical risk to satisfy CF, to the representation R = h(X, A; M, s), to the unconstrained ERM problem + demonstration that it suffices to learn parameters w is a bit convoluted and was hard (for me personally) to follow, despite the outline in Section 3. Please at least add an equation number for the argmin equation in Definition 1.

- Example 1 was not very easy for me to follow and did not help me build a greater intuition for your approach. It's possible that the notation-heavy nature made it more cumbersome to parse.

- I was unconvinced of the proof of Theorem 1 in the main paper, especially given that there was no mention of additional details in the Appendix. It's very important to indicate if any additional details/proofs/etc are included in the Appendix so the reader knows to reference that (otherwise, there's no guarantee that the reader will check)! Please add these references to related Appendix sections in all relevant places in the main paper.

- Nit: please use the citation reference style where the first author's name and year is displayed in place of a number. It makes it much easier to recall which reference corresponds to which paper.

- Nit: please use the full name for the "CVAE" and "DCEVAE" methods before abbreviating. Getting a sense of what these refer to without including the full name requires many more steps for the reader.

- Nit: Paper may benefit from additional peer read-through(s) before submitting for clarity, flow, catching typos.

**Questions:**

1. How does your method perform on different types of features, i.e. continuous, discrete, categorical, ordinal? Are there any data requirements?

2. What are the requirements for using your method in practice? Any hyperparameters or other dependencies? This information should be included for reproducibility. (Also, it's important to include a mention that reproducibility details are included in the Appendix!)

3. It's unclear from your paper what the role or benefits are from using a symmetric function-based representation is within the ERM problem you outline in Theorem 1, or what the motivation is for this choice of representation. Can you please provide this? (This is also missing from the main paper and is a suggestion to add.)

4. What is the sampling complexity for step 1 in Algo 1? Is it reasonable in practice to assume that the distribution calculated based on causal model M is known (referring to subscript 1 on page 3)? [I noted at the end of the paper your mention that finding a causal model is challenging - I think there's room to expand upon this, perhaps in the appendix.]

5. Your experiments used logistic regression and linear regression as base models - how does this approach perform with more complex tasks / base models, especially non-linear ones?

6. Do you plan to release any code?

**Limitations:**

The authors provide a brief acknowledgement of potential social impact (namely that counterfactual fairness is not the only notion one could consider and may not be the appropriate in some settings). Additionally, they briefly acknowledge that finding a causal model is challenging since there can be multiple causal models consistent with observational data.

---

> ### Author Rebuttal · Authors · 2023-08-09
>
> Thank you for your comment. Here are our responses to your questions one by one.
>
> Question 1: We do not have any requirements on the type of features. As you can see, we provided an algorithm that generates a new representation using the original feature and a counterfactual feature with a symmetric function. As long as the feature can be expressed in a numeric way, our algorithm worked. Note that categorical features can be represented numerically by one-hot encoding. Ordinal features also can be represented numerically by ordinal encoding.  In our experiments, the Law School dataset is composed of continuous features, and the UCI dataset consists of categorical features.
>
> Question 2: There is no special requirement for using our method. We included the hyper-parameters in the appendix. There are Hyper-parameters for the VAE (which was used to approximate the causal model). After training the VAE and approximating the causal model, generating counterfactual fair representation does not need any hyper-parameter tuning for a given symmetric function $s$ (symmetric function $s$ has been used in algorithm 1 and algorithm 2).
>
> Question 3: Our motivation for using such a symmetric function comes from the definition of counterfactual fairness (Definition 1 in the paper). Counterfactual fairness implies that the decision should be the same in the factual and counterfactual world. Algorithms 1 and 2 find a presentation that is a symmetric function of the factual and counterfactual world. This means that if replace the counterfactual world with the factual world, the presentation still remains the same and does not change. As a result, representation $R$ and the downstream classifier/regressor remain counterfactually fair. Without such a symmetric function, it is hard to find a representation under counterfactual fairness.
>
> Question 4: In our experimental study, we use VAE to sample $u$. The time complexity of generating $u$ is $\mathcal{O}$(#of parameters of VAE). We can use other algorithms to generate $u$. For example,  MCMC is another algorithm that has been used by [22] and its time complexity has been discussed in several works (see e.g., https://arxiv.org/pdf/0704.2167.pdf)
>
> Regarding the causal model, we assume that the causal model is known. This is the common assumption in the causal fairness literature (e.g. [22], which is the original paper that introduces counterfactual fairness). Once the causal structure is known, the structural equations can be estimated using VAEs. In the appendix, we explained how VAE can be used. If the causal model is not known,  human knowledge and causal structure learning methods (see e.g., https://arxiv.org/pdf/1706.09141.pdf) can help to find the causal structure. The general problem of learning a causal structure is beyond the scope of this paper.
>
> Question 5: Our method is able to generate a representation that will be used as the input of a machine-learning model.  Our algorithm generates a counterfactually fair representation regardless of the downstream task. The representation is independent of the machine learning model used in the downstream task.  Nevertheless, the followings are the numerical results with MLP (with default in sklearn). Our method still gets an accuracy of 79.55% and a TE of 0.0256. In this case, the CE method achieves 79.33% accuracy.
>
> | Method | Accuracy            | TE                  | TE0                 | TE1                 |
> |--------|---------------------|---------------------|---------------------|---------------------|
> | UF     | $0.8199 \pm 0.0013$ | $0.1494 \pm 0.0103$ | $0.1187 \pm 0.014$  | $0.1644 \pm 0.0103$ |
> | CA     | $0.8142 \pm 0.0027$ | $0.1838 \pm 0.0103$ | $0.1312 \pm 0.0050$ | $0.2098 \pm 0.0151$ |
> | CE     | $0.7933 \pm 0.0010$ | 0                   | 0                   | 0                   |
> | Ours   | $0.7955 \pm 0.0017$ | $0.0256 \pm 0.0031$ | $0.0212 \pm 0.0043$ | $0.0278 \pm 0.0037$ |
>
> Question 6: Yes, we plan to release the code. Here is an anonymous link to the code (https://anonymous.4open.science/r/CF_Representation-B01B/readme.md).
>
> And thank you for your comment about the writing. We take your comment seriously and will write it in a more organized way in the camera-ready version.

---

### Decision · Program_Chairs · 2023-09-21

**Decision:**

Accept (poster)

**Comment:**

This paper proposes a novel approach for generating counterfactually fair representations through the use of a symmetric function. The authors then show that any downstream models trained with such representations can achieve perfect/exact CF. In the initial reviews, the reviewers had concerns about the choice of metrics used in the experiments, the presentation of the numerical results, as well as the presentation of the overall paper. Some of the comments have been partially addressed in the rebuttal. The presentation needs further improvement.